**Data Availability Statement:** All National HIV Surveillance System (NHSS) data, as used in this analysis, are publicly available. NHSS data are periodically updated and in its most recent form

# Impact of viral suppression among persons with HIV upon estimated HIV incidence between 2010 and 2015 in the United States

**Taraz Samandari** *, **Jeffrey Wiener, Ya-Lin A. Huang, Karen W. Hoover, Azfar-e-Alam Siddiqi**

Division of HIV/AIDS Prevention, Centers for Disease Control & Prevention, Atlanta, Georgia, United States of America

* tts0@cdc.gov

## Abstract

### Background

The suppression of viremia among persons with HIV (PWH) using antiretroviral therapy has been hypothesized to reduce HIV incidence at the population level. We investigated the impact of state level viral suppression among PWH in the United States on estimated HIV incidence between 2010 and 2015.

### Methods

Viral suppression data and HIV incidence estimates from the National HIV Surveillance System were available from 29 states and the District of Columbia. We assumed a one year delay for viral suppression to impact incidence. Poisson regression models were used to calculate the estimated annual percent change (EAPC) in incidence rate. We employed a multivariable mixed-effects Poisson regression model to assess the effects of state level race/ethnicity, socioeconomic status, percent men who have sex with men (MSM) and hepatitis C virus prevalence as a proxy for injection drug use on HIV incidence.

### Findings

Fitted HIV incidence for 30 jurisdictions declined from 11.5 in 2010 to 10.0 per 100,000 population by 2015 corresponding with an EAPC of -2.67 (95% confidence interval [95%CI] -2.95, -2.38). Southern states experienced the highest estimated incidence by far throughout this period but upon adjustment for viral suppression and demographics there was a 36% lower incidence rate than Northeast states (adjusted rate ratio [aRR] 0.64; 95%CI 0.42, 0.99). For every 10 percentage point (pp) increase in viral suppression there was an adjusted 4% decline in HIV incidence rate in the subsequent year (aRR 0.96; 95%CI 0.93, 0.99). While controlling for viral suppression, HIV incidence rate increased by 42% (aRR 1.42 95%CI 1.31, 1.54) for every 5 pp increase in percent Black race and by 27% (aRR 1.27 95%CI 1.10, 1.48) for every 1 pp increase in percent MSM in states.

are available at the following URLs https://www.cdc.gov/nchhstp/atlas/index.htm and https://www.cdc.gov/hiv/library/reports/hiv-surveillance.html. National Health and Nutrition Examination Survey used in this analysis can be accessed here: https://wwwn.cdc.gov/nchs/nhanes/Default.aspx. US Census data is downloadable from: https://www.census.gov/data/tables/time-series/demo/popest/2010s-state-total.html.

**Funding:** The author(s) received no specific funding for this work.

**Competing interests:** The authors have declared that no competing interests exist.

## Interpretation

A decline in estimated HIV incidence from 2010 to 2015 was associated with increasing viral suppression in the United States. Race and sexual orientation were important HIV acquisition risk factors.

## Introduction

Highly active antiretroviral therapy–subsequently referred to as antiretroviral therapy (ART)–was introduced in the United States in 1996. Shortly thereafter the United States Centers for Disease Control and Prevention (CDC) reported the first substantial decline in AIDS deaths [1]. Health economists determined that by 2003 at least three million life years had been saved in the United States as a direct result of ART [2].

Treatment of HIV-infected persons to prevent the onward transmission of the virus to uninfected persons has been the recommendation of public health officials in the United States since 2012. Viewed from the perspective of public health prevention, this strategy is sometimes referred to as 'treatment as prevention' and has contributed to the announcement of a new initiative by the United States government namely 'Ending the HIV Epidemic' [3]. A treatment as prevention approach was supported by the early observation that in the island of Taiwan, after ART was provided freely to a population of approximately 4,300 persons with HIV (PWH), the transmission rate of HIV declined [4]. Ample proof of the transmission-blocking potential of ART between sero-discordant couples was later published in 2011 as the chief outcome of a multi-country randomized clinical trial [5]. Subsequent analysis of this trial showed that almost all transmissions to uninfected partners in this trial occurred from HIV-infected participants whose last index viral load before the estimated date of infection of their uninfected partners exceeded 40,000 copies/mL [6].

Similar associations between reduced viral load and reduced HIV incidence were observed in British Columbia, Canada, and in the context of a community trial in India. In British Columbia, a population-level longitudinal analysis of province-wide registries showed a 1% decline in HIV incidence for every 1% increase in viral suppression [7]. The authors concluded that ART expansion between 1996 and 2012 was associated with "a sustained and profound population-level decrease in morbidity, mortality and HIV transmission." In India, a cluster-randomized trial was conducted among more than 26,000 injection drug users and men who have sex with men (MSM) from 22 communities. These researchers concluded that to reduce HIV incidence by 1%, the viremia in the population would have to be reduced by 4.3% and ART use by PWH would have to increase by 19.5% [8].

In the United States, a Poisson model using "community viral load" in the city of San Francisco showed that an increase in viral suppression from 45% to 78% was associated with a 46% decline in diagnoses of HIV between 2004 and 2008 [9]. However, the impact of viral suppression on incident HIV has not been previously reported for the United States at the state, regional or national levels.

Over the past decade, jurisdictions have increasingly provided reports of viral load data from persons living with diagnosed HIV and CD4+ lymphocyte counts of newly diagnosed cases to the CDC. The latter has allowed CDC to estimate incident HIV by jurisdiction. In this paper, we examined the impact of viral suppression on estimated incident HIV by jurisdiction in the six-year period from 2010 to 2015.

## Methods

The National HIV Surveillance System (NHSS) serves as the source for both HIV diagnosis and viral suppression data. CDC estimates and publishes HIV incidence using the reported diagnosis data following the methodology of Song et al. using a CD4 depletion model to estimate time between infection and diagnosis [10, 11]. The estimation method also makes use of first reported CD4 cell count after HIV diagnosis, assuming that persons are ART naïve at the time. Reporting of first CD4 test result after diagnosis is a required element in NHSS and significant progress has been made in reporting of all levels and percentages of CD4 test results [12]. For this analysis, we used CDC's published HIV incidence estimates among persons 13 years or older (adults and adolescents) for the years 2010 through 2015 [13].

Relative standard errors (RSEs) were previously calculated for estimates of HIV incidence per state. Estimates with an RSE > 50% are considered statistically unreliable and ordinarily not published, and were not included in this analysis. To understand the potential impact of this source of missing data, a sensitivity analysis was done by including these suppressed incidence estimates in the final statistical model to evaluate the change in fitted incidence rates.

We included 2010–2015 state-level demographic variables in the analysis, including percent less than high school graduate, percent Black/African American, percent Hispanic, percent uninsured, and percent below poverty level. These demographic data were extracted from the American Community Survey and the Current Population Survey [14, 15]. Approximately 60% of PWH in the United States are gay, bisexual, or other MSM. We included published estimates of percentage of MSM among adult men for each state [16]. Recent increases in injection drug use in the United States has resulted in growing hepatitis C (HCV) and HIV infections in some rural areas. In this study, we also included prevalence of HCV infection by state in the analysis as a proxy for the prevalence of injection drug use using published estimates based on National Health and Nutrition Examination Survey 2009–2013 [17].

Viral suppression among persons living with HIV (diagnosed and undiagnosed) was extracted from various supplemental surveillance reports published by CDC and is freely available online. The percent virally suppressed was calculated using the following sets of numbers. The numerator was the number of persons with suppressed viral load test results reported to CDC from jurisdictions with complete laboratory reporting of CD4 and viral load test results. Standard surveillance definition of <200 copies/mL at the most recent test result in each year, was used to define viral suppression in a given calendar year. The denominator consisted of estimated total prevalence of HIV for the same year, count of persons living with diagnosed HIV plus estimate of persons living with undiagnosed HIV. The NHSS data is updated regularly. The most recent numerator and denominator data can be accessed from either the HIV surveillance supplemental reports [12, 13]. or the National Center for HIV/AIDS Viral Hepatitis STD and Tuberculosis Prevention's AtlasPlus, available at https://www.cdc.gov/nchhstp/atlas/index.htm. To assess the impact of viral suppression on estimated HIV incidence between 2011 and 2015, a one-year lag in viral suppression was used thereby restricting the viral suppression data in our analysis to the 2010–2014 time frame. Although viral suppression data were available for 39 jurisdictions for at least one year, we restricted the analysis to 30 jurisdictions (29 states and the District of Columbia) with data available for 60% (3 years or more) of the time frame to ensure evaluation of changes over time.

Means and standard deviations were used to summarize percent viral suppression and state level demographic variables by year. Quartiles for estimated HIV incidence rate and percent viral suppression were calculated across the entire time frame, and used to evaluate changes in these measures over time. Estimated annual percent changes (EAPC) in HIV incidence with corresponding fitted yearly rates were determined using Poisson regression models that

included a fixed effect for year for each jurisdiction and for all 30 jurisdictions combined. Fitted yearly incidence rates were also determined by region (Northeast, Midwest, South, and West). Unadjusted and adjusted rate ratios associated with differences in percent viral suppression and demographic characteristics were calculated using multivariable mixed-effects Poisson regression models that included a fixed effect for year, and a random intercept to account for within-state correlations. Percent viral suppression and state level demographics were treated as continuous covariates and adjusted models included 2010 estimated HIV incidence rate as a covariate. All analyses were conducted using SAS version 9.4 and R version 3.5.0.

## Results

Of the 30 United States jurisdictions included in the analysis, one third were located in the Midwest and one third were in the South (Table 1). State specific fitted annual HIV incidence rates showed that 17 of 30 jurisdictions experienced negative estimated annual percentage change (EAPC) between 2010 and 2015 including three of the most populous states: California, New York and Texas (Table 2). In 2010, Washington DC had the highest fitted incidence of any jurisdiction during this period at 149.7 per 100,000 population but also the greatest estimated annual percent decline, -12.7 (95% confidence intervals [CI] -15.0, -10.3). Seven states had positive EAPC with Indiana and Hawaii experiencing the largest increases, 9.1 (95% CI 6.0, 12.3) and 14.0 (95% CI 6.7, 21.8) respectively.

**Table 1. Estimated HIV incidence rate, state level viral suppression and demographics by year for 29 states and the District of Columbia[*].**

| | 2010 | 2011 | 2012 | 2013 | 2014 | 2015 | Percent Change from 2010 to 2015 |
|---|---|---|---|---|---|---|---|
| Estimated HIV Incidence Rate per 100,000 | 17.1 | 16.6 | 16.1 | 15.6 | 15.7 | 15.6 | -8.7 |
| | Mean (standard deviation) | | | | | | |
| Percent Viral Suppression | 32.3 (6.6) | 36.2 (5.9) | 40.2 (6.4) | 45.4 (7.6) | 47.4 (7.6) | 49.4 (7.1) | 53.0 |
| Percent Less Than High School Graduate | 10.9 (3.4) | 10.6 (3.3) | 10.1 (3.2) | 9.8 (3.1) | 9.5 (3.0) | 9.3 (3.0) | -15.0 |
| Percent Black | 11.9 (12.6) | 11.8 (12.5) | 11.9 (12.3) | 11.9 (12.3) | 12.0 (12.3) | 11.9 (12.1) | 0.3 |
| Percent Hispanic | 9.0 (8.8) | 9.1 (8.9) | 9.3 (8.9) | 9.4 (8.9) | 9.6 (9.0) | 9.8 (9.0) | 9.3 |
| Percent Uninsured | 13.8 (3.9) | 13.6 (3.9) | 13.2 (4.0) | 13.0 (3.6) | 10.7 (3.5) | 8.8 (3.3) | -36.0 |
| Percent Below Poverty Level | 13.9 (3.5) | 14.0 (3.4) | 13.7 (3.3) | 14.3 (4.0) | 13.5 (3.8) | 12.3 (2.9) | -11.5 |
| | Mean (standard deviation) | | | | | | |
| Percent MSM 2009–13[¶] | 3.5 (2.5) | | | | | | |
| Prevalence of HCV Infection 2013–16[¶] | 0.9 (0.4) | | | | | | |
| Region†[¶] | | | | | | | |
| Northeast | 10.0% (3 states) | | | | | | |
| Midwest | 33.3% (10 states) | | | | | | |
| South | 33.3% (9 states + DC) | | | | | | |
| West | 23.3% (7 states) | | | | | | |

[*] States were included in the subset if viral suppression data was available for 60% of study years, 2010–14.

† States included in the subset by region: Northeast (Maine, New Hampshire, New York); Midwest (Illinois, Indiana, Iowa, Michigan, Minnesota, Missouri, Nebraska, North Dakota, South Dakota, Wisconsin); South (Alabama, District of Columbia [DC], Georgia, Louisiana, Maryland, South Carolina, Tennessee, Texas, Virginia, West Virginia); West (Alaska, California, Hawaii, Oregon, Utah, Washington, Wyoming).

¶ Percent MSM and prevalence of HCV infection were estimated from the noted data sources over the entire timeframe indicated, not separately by year.

Note. Data sources include: National HIV Surveillance System (estimated HIV incidence and viral suppression for persons with HIV); American Community Survey 2010–15 (percent less than high school graduate, Black, Hispanic, uninsured); Current Population Survey 2010–15 (percent below poverty level); published estimates using data from the American Community Survey 2009–2013 (percent men who have sex with men [MSM]) [15]; National Health and Nutrition Examination Survey 2013–16 (prevalence of HCV infection).

**Table 2. Estimated annual percent change (EAPC) in HIV incidence rate by state with corresponding fitted annual incidence rates per 100,000.**

| Fitted incidence rate | 2010 | 2011 | 2012 | 2013 | 2014 | 2015 | EAPC (95% CI)* |
|---|---|---|---|---|---|---|---|
| 30 Jurisdictions | 11.5 | 11.2 | 10.9 | 10.6 | 10.3 | 10.0 | -2.67 (-2.95, -2.38) |
| Northeast: | | | | | | | |
| Maine¶ | - | - | - | - | - | - | - |
| New Hampshire¶ | - | - | - | - | - | - | - |
| New York | 26.1 | 24.7 | 23.4 | 22.1 | 20.9 | 19.8 | -5.38 (-6.40, -4.35) |
| Midwest: | | | | | | | |
| Illinois | 14.2 | 14.0 | 13.7 | 13.4 | 13.1 | 12.9 | -1.98 (-3.67, -0.25) |
| Indiana | 7.8 | 8.6 | 9.3 | 10.2 | 11.1 | 12.1 | 9.13 (6.01, 12.33) |
| Iowa† | 3.6 | 3.7 | 3.9 | 4.1 | - | - | 4.69 (-13.87, 27.24) |
| Michigan | 8.9 | 8.8 | 8.8 | 8.7 | 8.7 | 8.6 | -0.59 (-3.00, 1.89) |
| Minnesota† | 7.5 | 7.2 | 6.9 | 6.6 | 6.3 | 6.1 | -4.17 (-7.76, -0.43) |
| Missouri | 10.7 | 10.1 | 9.5 | 8.9 | 8.4 | 7.9 | -5.85 (-8.71, -2.89) |
| Nebraska† | 6.5 | 5.7 | 5.1 | - | - | - | -11.56 (-66.42, 132.89) |
| North Dakota¶ | - | - | - | - | - | - | - |
| South Dakota¶ | - | - | - | - | - | - | - |
| Wisconsin† | 5.1 | 5.1 | 5.1 | 5.0 | 5.0 | 4.9 | -0.81 (-4.96, 3.52) |
| South: | | | | | | | |
| Alabama | 15.7 | 15.1 | 14.6 | 14.1 | 13.6 | 13.1 | -3.62 (-6.25, -0.92) |
| District of Columbia | 149.7 | 130.7 | 114.1 | 99.6 | 87.0 | 75.9 | -12.69 (-15.05, -10.27) |
| Georgia | 30.9 | 30.0 | 29.0 | 28.1 | 27.3 | 26.4 | -3.11 (-4.44, -1.77) |
| Louisiana | 25.6 | 25.7 | 25.9 | 26.1 | 26.2 | 26.4 | 0.65 (-1.46, 2.79) |
| Maryland | 30.0 | 28.3 | 26.7 | 25.1 | 23.7 | 22.3 | -5.77 (-7.50, -4.00) |
| South Carolina | 18.9 | 18.0 | 17.2 | 16.4 | 15.7 | 14.9 | -4.55 (-6.97, -2.07) |
| Tennessee | 15.9 | 15.5 | 15.0 | 14.6 | 14.2 | 13.7 | -2.92 (-5.17, -0.61) |
| Texas | 21.8 | 21.3 | 20.8 | 20.3 | 19.8 | 19.4 | -2.28 (-3.26, -1.29) |
| Virginia | 13.0 | 13.1 | 13.2 | 13.2 | 13.3 | 13.4 | 0.61 (-1.58, 2.86) |
| West Virginia† | 4.2 | 4.4 | 4.7 | 5.0 | 5.3 | - | 6.26 (-5.94, 20.04) |
| West: | | | | | | | |
| Alaska¶ | - | - | - | - | - | - | - |
| California | 16.6 | 16.4 | 16.2 | 16.1 | 15.9 | 15.7 | -1.06 (-1.97, -0.13) |
| Hawaii† | 6.2 | 7.0 | 8.0 | 9.2 | 10.4 | 11.9 | 13.99 (6.66, 21.83) |
| Oregon† | 7.6 | 7.1 | 6.7 | 6.3 | 5.9 | 5.5 | -6.10 (-10.26, -1.74) |
| Utah† | 5.0 | 5.1 | 5.3 | 5.4 | 5.5 | 5.6 | 2.31 (-3.75, 8.75) |
| Washington | 9.2 | 8.8 | 8.4 | 8.0 | 7.6 | 7.3 | -4.74 (-7.60, -1.79) |
| Wyoming¶ | - | - | - | - | - | - | - |

* EAPC and fitted yearly HIV incidence rates for each state determined from separate Poisson regression models including a fixed effect for year.

† One or more years of HIV incidence estimates for these states had relative standard errors of 30%-50%.

¶ EAPC could not be estimated if a state had <3 years of available HIV incidence estimates.

Fitted HIV incidence rates determined from a mixed effects Poisson regression model showed that all four United States regions experienced declines in HIV incidence from 2010 to 2015 (Fig 1). The Southern states experienced the greatest decline but also retained–by far–the highest fitted annual rate at 18.65 per 100,000 by 2015.

Comparing 2015 with 2010, most states and DC either began reporting viral suppression data or had improvements in the proportion of PWH virally suppressed (Fig 2). Most states did not experience changes to a lower quartile of estimated HIV incidence rate, although one populous state (New York) did, as did two others (Minnesota and Oregon). Overall mean

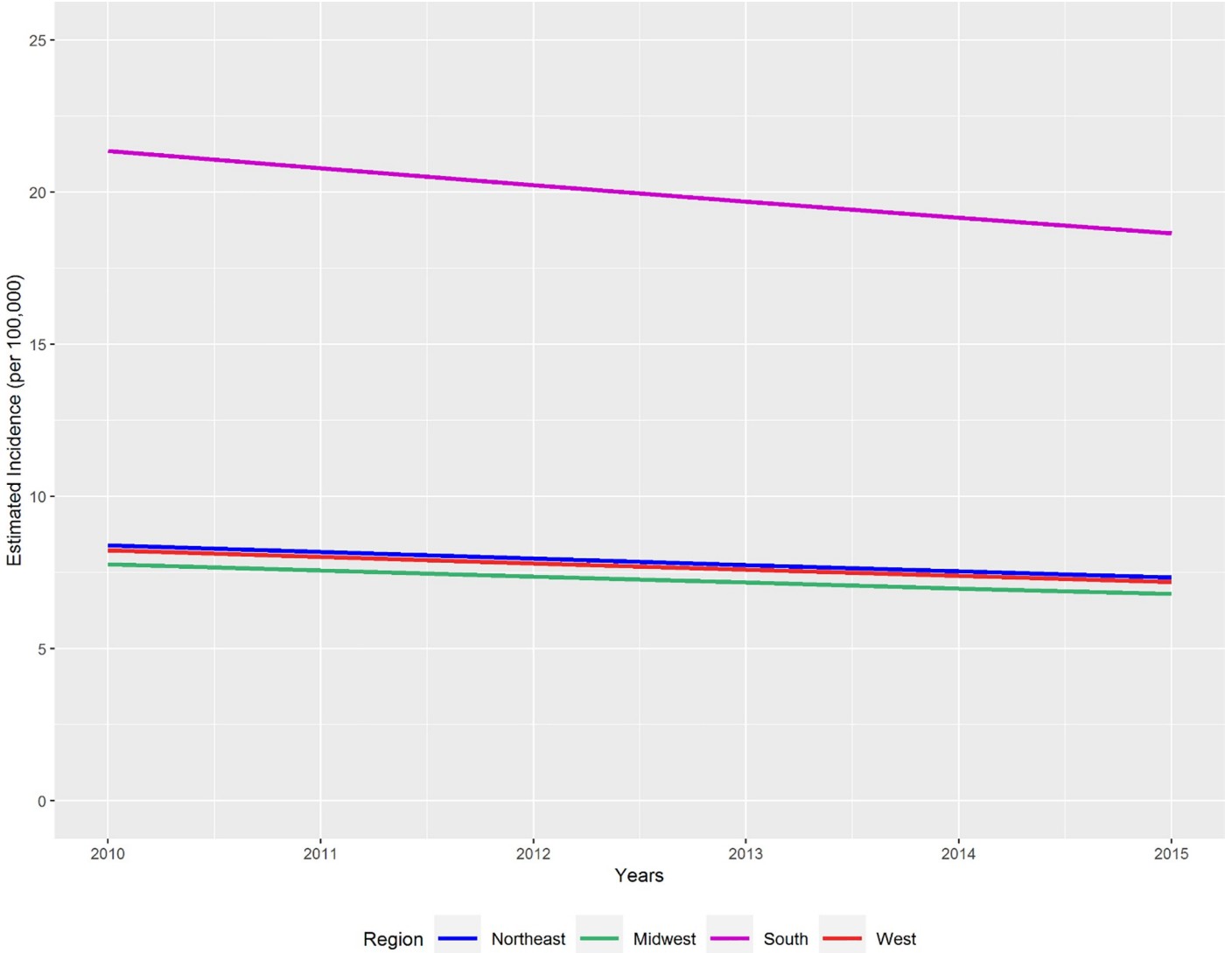

**Fig 1. Fitted HIV incidence rate by region, United States 2010–2015.** Fitted HIV incidence rate determined using a mixed-effects Poisson regression model including region, a fixed effect for year, and a random intercept.

(±standard deviation) viral suppression among PWH steadily increased from 32.3% ± 6.6% in 2010 to 49.4% ± 7.1% in 2015 (Table 1) for the 30 jurisdictions.

After adjustment for proportions of persons with less than a high school education, Black race, Hispanic ethnicity, below the poverty line, uninsured, MSM, HCV prevalence and virally suppressed PWH, the multivariable model showed that Northeastern states had the highest fitted HIV incidence rate between 2011 and 2015 (Fig 3). The high adjusted incidence rates observed in the Northeast in this analysis contrast with unadjusted incidence rates observed in the South (Fig 1).

In the multivariable model, we found a 4% decline in estimated HIV incidence rate for every 10 percentage point (pp) increase in viral suppression (adjusted rate ratio [aRR] = 0.96; 95% confidence interval (CI) = 0.93, 0.99, Table 3). While adjusting for viral suppression and relevant state level demographic characteristics, every 5 pp increase in the proportion of persons with less than a high school education, of Black race and below the poverty line, higher

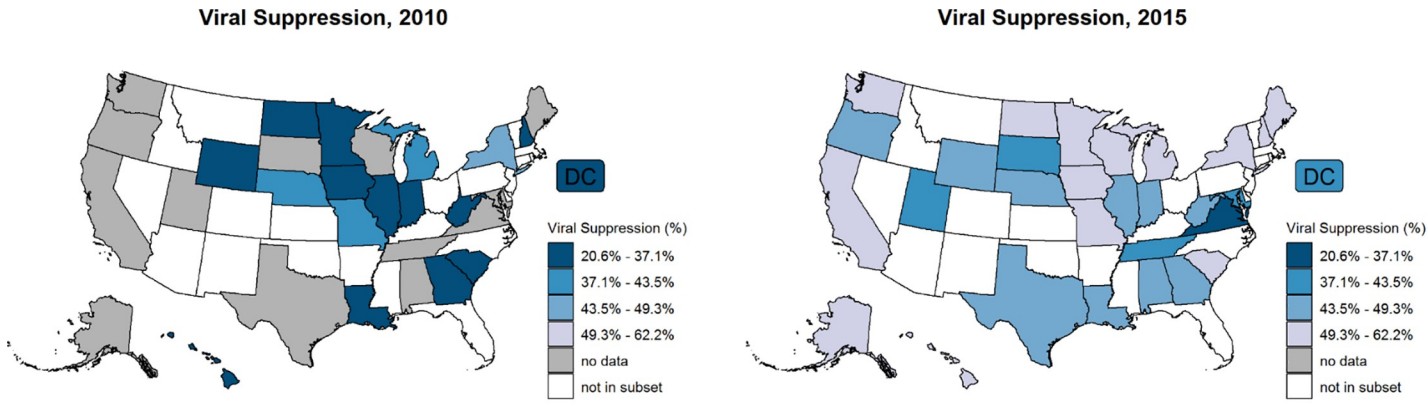

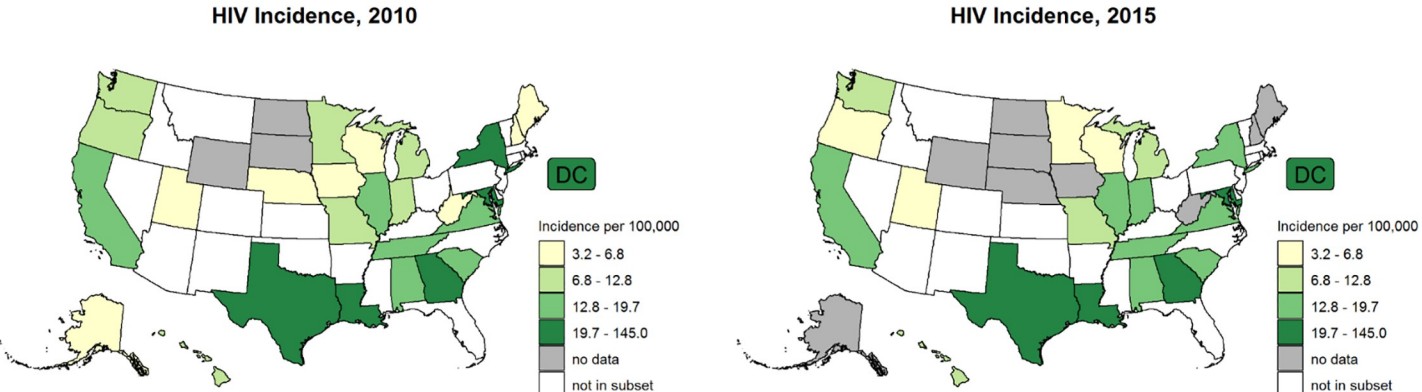

**Fig 2. Percentage viral suppression in persons with HIV and estimated HIV incidence in 2010 and 2015, United States.** The subset includes 30 jurisdictions (29 states and the District of Columbia) with viral suppression data available for 3 years or more. Estimated HIV incidence rate and percent viral suppression categories were determined by quartiles calculated across the entire time frame 2010–15.

HIV incidence rates were observed: (aRR 1.20, 95% CI 1.04, 1.38; aRR 1.42, 95% CI 1.31, 1.54; aRR 1.10, 95% CI 1.07, 1.14; respectively). For every 1 pp increase in adult men who were MSM, there was a 27% increase in HIV incidence rate (aRR 1.27, 95% CI 1.10, 1.48). We found a 5% (aRR 0.95; 95% CI 0.92, 0.97) decline in HIV incidence rate for every 5 pp increase in percent uninsured.

The largest impact of viral suppression after adjusting for relevant state level demographic characteristics was found in the Southern states: aRR 0.64 (95% CI 0.42, 0.99) as compared with the Northeastern states (Table 3). Impacts were slightly less in the Midwest: aRR 0.70 (95% CI 0.50, 0.99) and the West: aRR 0.72 (95% CI 0.49, 1.06). Fig 3 suggests that if percent viral suppression increased from 30% to 50%, the fitted incidence would decrease from 18.8 to 17.3 per 100,000 in the Northeast in 2015, while it would decrease from 12.1 to 11.1 per 100,000 in the South.

Although HCV prevalence was strongly associated with a RR of 2.83 in the unadjusted model, it was not upon adjustment (aRR 1.00), suggesting that trends in injection drug use (for which HCV prevalence serves as a proxy) during these years was not an important independent driver of HIV incidence.

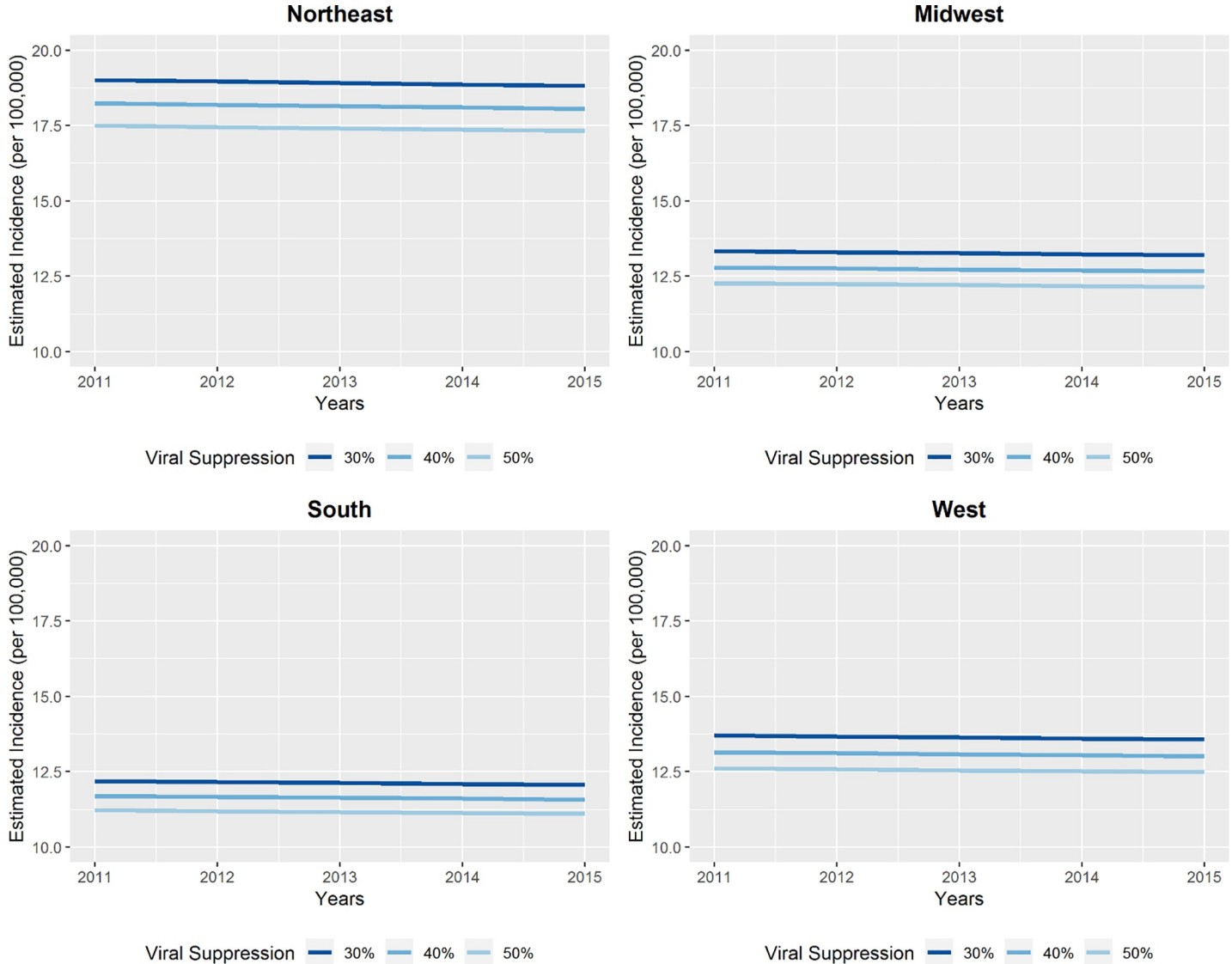

**Fig 3. Fitted HIV incidence rate by viral suppression in previous year (%) and U.S. region, 2011–2015.** Fitted HIV incidence rate determined using a multivariable mixed-effects Poisson regression model including all variables in Table 3, fixed effects for year and 2010 HIV incidence rate and a random intercept.

The sensitivity of this model to missing data was evaluated by including HIV incidence estimates that are ordinarily suppressed in the surveillance system due to being statistically unreliable with relative standard error >50%. No changes in fitted incidence rates larger than 10% were found.

## Discussion

We observed that between 2010 and 2015 among 29 states and the District of Columbia every 10 pp increase in viral suppression during the previous year was associated with an adjusted 4% decline in the estimated incidence rate of HIV in the subsequent year. This statistical association is consistent with the magnitude of the effects observed in community-randomized trials conducted in India, the SEARCH trial and the PopART trial, as well as the public health experience of British Columbia [7, 8, 18]. By scaling the statistics of published data, these

**Table 3. Unadjusted and adjusted associations of estimated HIV incidence rate with state level viral suppression and demographic variables.**

| | Unadjusted Rate Ratio* | | Adjusted Rate Ratio† | |
|---|---|---|---|---|
| | (95% CI) | p-value | (95% CI) | p-value |
| Viral Suppression (%) in previous year (per increase of 10 pp) | 1.02 (0.99, 1.05) | 0.117 | 0.96 (0.93, 0.99) | 0.005 |
| Percent < High School Graduate (per increase of 5 pp) | 1.07 (0.95, 1.21) | 0.243 | 1.20 (1.04, 1.38) | 0.014 |
| Percent Black (per increase of 5 pp) | 1.34 (1.27, 1.42) | < 0.001 | 1.42 (1.31, 1.54) | < 0.001 |
| Percent Hispanic (per increase of 5 pp) | 1.04 (0.89, 1.21) | 0.642 | 1.06 (0.99, 1.13) | 0.091 |
| Percent Uninsured (per increase of 5 pp) | 0.93 (0.91, 0.95) | < 0.001 | 0.95 (0.92, 0.97) | < 0.001 |
| Percent Below Poverty Level (per increase of 5 pp) | 1.05 (1.02, 1.08) | 0.002 | 1.10 (1.07, 1.14) | < 0.001 |
| Percent MSM 2009–13 (per increase of 1 pp) | 1.22 (1.12, 1.33) | < 0.001 | 1.27 (1.10, 1.48) | 0.002 |
| Prevalence of HCV Infection 2013–16 (per increase of 1 pp) | 2.83 (1.58, 5.09) | < 0.001 | 1.00 (0.69, 1.44) | 0.995 |
| Region | | | | |
| Northeast | 1.0 | | 1.0 | |
| Midwest | 0.90 (0.38, 2.11) | 0.808 | 0.70 (0.50, 0.99) | 0.048 |
| South | 2.48 (1.08, 5.69) | 0.032 | 0.64 (0.42, 0.99) | 0.045 |
| West | 1.05 (0.42, 2.63) | 0.915 | 0.72 (0.49, 1.06) | 0.099 |

* Results determined using a multivariable mixed-effects Poisson regression models including the variable of interest, a fixed effect for year, and a random intercept.

† Results determined using a multivariable mixed-effects Poisson regression model including all variables in the table, fixed effects for year and 2010 HIV incidence rate, and a random intercept.

pp = percentage point.

studies observed a 2.3%, 6.7%, 5.4% and a 10% decline in HIV incidence rate for every 10 pp increase in viral suppression, respectively. Our ecologic analysis is remarkable considering that the populations of these 30 jurisdictions amount to 64.5% of the United States population in 2010, and are not only diverse in income, race, and ethnicity but are also subject to diverse health insurance contexts.

By virtue of having the highest incidence of any United States region, Southern states also have the opportunity to make the greatest impact on incident HIV with better viral suppression of PWH. In 2014, Southern states account for an estimated 44% percent of all PWH (diagnosed and undiagnosed) in the United States despite being populated by only a third (37%) of the overall population. Geographically, Southern states are a focus region for the 'Ending the HIV Epidemic' initiative.

Notable national demographic changes during the six-year period analyzed included a 15.0% decline in the proportion with less than a high school education, an 11.5% decline in the proportion of the United States population below the poverty level and a 36.0% decline in the percentage uninsured, particularly in 2014 and 2015. After adjustment for viral suppression and demographic characteristics, HIV incidence rate was 36% lower in Southern states compared to northeastern states. While health insurance coverage tends to be better in Northeastern states than Southern states [19], an analysis of private health insurance billing data between 2012 and 2013 found that the percentage of persons without an antiretroviral

prescription was highest for persons residing in the Northeast region at 30.8% [20]. This observation suggests a possible link between reduced viral suppression among privately insured diagnosed PWH and HIV incidence in the Northeast.

In multivariable analysis, state level percent Black race had the strongest association with the estimated HIV incidence rate followed by percent MSM. Blacks/African Americans endure a disproportionate burden of new HIV diagnoses in the United States: in 2017, they accounted for 13% of the United States population but 43% of new HIV diagnoses. This racial disparity in HIV diagnoses has been linked to poverty, substandard education, unstable housing, inadequate health insurance, disproportionate incarceration rates and limited social mobility [21–23]. Additionally, gay and bisexual men accounted for 67% of all HIV diagnoses in 2016 despite being approximately 4% of the United States male population [16] and, of these new diagnoses in MSM, 38% were among Black/African American persons. Rather than geographic region being the explanatory variable, confounding state level demographic variables appear to explain the disproportionate burden of HIV incidence in Northeastern states.

We observed a trend of overall increases in viral suppression and decreased HIV incidence across the United States. In 2014, the major provisions of the Affordable Care Act (ACA) of 2010 –which included an expansion of Medicaid (the federal insurance program for persons requiring health insurance)–came into force. This expansion played a vital role in increasing access to and provision of care and treatment for PWH. However, not all states expanded Medicaid and some essential HIV services and medications were not covered by Medicaid. Typically, these outstanding services (e.g., case management, transportation, pharmacy services) were provided by the Ryan White HIV/AIDS Program (RWHAP), a federal program established in 1990 to fill gaps in HIV care and treatment for low-income PWH [24]. By 2016, the uninsured share of the United States population had roughly halved, with estimates ranging from 20 to 24 million additional people covered. The decline in the uninsured population was captured in our analysis.

Given this context, one may examine the interplay we observed between HIV incidence, insurance and poverty. While controlling for percent viral suppression, our analysis showed a 10% increase in the HIV incidence rate for every 5 pp increase in the percent of persons living below the national poverty level, but that for every 5 pp increase in the percent of uninsured persons was associated with a 5% decrease in the HIV incidence rate. Superficially, these two observations may appear contradictory because poverty would appear to correlate with lack of insurance. Indeed, nearly half of all PWH in the United States have a household income at or below the federal poverty line and 28.4% of PWH had no insurance [25]. However, a recent analysis of the RWHAP showed that, among recipients, viral suppression increased from 69.5% in 2010 to 85.9% in 2017 [26]. The RWHAP could explain the association we observed of a decreased HIV incidence rate among the uninsured. A separate analysis of the period 2009–2013 showed that, among 18,095 PWH with and without RWHAP assistance, those who were uninsured and underinsured were more likely to receive ART and be virally suppressed than those with other types of healthcare coverage [27].

Indiana and West Virginia had increases in incident HIV over time, and were also among the states most impacted by the growing opioid and injection drug use epidemics [28]. We therefore attempted to assess whether injection drug use impacted the relationship between viral suppression and HIV incidence. We included HCV prevalence by jurisdiction in our model as a proxy for injection drug use because HCV infection is strongly associated with injection drug use. We did not observe an association between HCV prevalence and HIV incidence rate after adjusting for relevant demographic characteristics. This observation is consistent with surveillance reports from the same period that show that injection drug use was a relatively low contributor to the annual number of new diagnoses.

Our analysis has a number of limitations, chief among them being that by 2015 not all states were reporting all viral load and CD4+ lymphocyte test results data. Even some of the 30 jurisdictions in our subset did not have viral suppression data for all years between 2010 and 2015, or had HIV incidence estimates that were statistically unreliable and therefore not included in the analysis. Among the states not included was Florida, which–after California and Texas–is the third most populous state. Nevertheless, the jurisdictions included in this analysis represent almost two thirds of the United States population. Another limitation is that we used a single measure of viral load to determine who was or was not virally suppressed; however single measures of viral load do not necessarily approximate durable viral suppression [29]. While some analyses examined the impact of ART coverage on HIV incidence [7], a strength of our analysis is the examination of the use of viral suppression to impact HIV incidence rather than ART coverage given the well-known phenomenon of nonadherence to ART [30]. Finally, our analysis does not account for changes in condom use or the use of pre-exposure HIV prophylaxis (PrEP). The increases of sexually transmitted diseases during this period in the United States–to the highest recorded levels–appear to confirm that condom use declined during this period [31–33]. CDC recommended PrEP in 2014 and by 2016 an estimated 78,000 people had filled a prescription for it [34]. By 2016 only 7% of the estimated 1.1 million U.S. persons who had indications for PrEP were prescribed it [35]. It is unlikely that PrEP-related decreases in HIV acquisition would have significantly affected our analysis.

A key area for improvement in order to more fully realize the strategy of treatment as prevention is recognizing that an estimated 43% of new HIV transmissions were generated by persons who were aware of their HIV infection but were not in care, and 38% of new infections were generated by persons unaware of their status [36]. Therefore, identifying persons with undiagnosed HIV and re-engaging and initiating ART for approximately half a million PWH who were not linked to care or who dropped out of care are vital to successfully prosecuting the 'Ending the HIV Epidemic' campaign in the United States.

In conclusion, our ecologic analysis suggests that between 2010 and 2015, the increase in viral suppression observed after the rollout of ART was associated with a modest reduction in the United States' estimated HIV incidence rate. Initiating and adhering to ART with regimens that include integrase inhibitors and single-tablet triple therapies have made viral suppression easier than ever before. Based upon another ecologic analysis that controlled for viral suppression, PrEP may already have made a contribution towards reducing incident HIV in the United States [37]. Our analysis appears to underscore the importance of the RWHAP in helping uninsured and under-insured PWH achieve viral suppression and thereby reduce incident HIV, and that the greatest impact of viral suppression may be effected in Southern states. The 'Ending the HIV Epidemic' announced as a national strategy in 2019 is aimed at intensifying treatment as prevention and PrEP in the 48 counties with the highest burden of HIV in order to reduce national HIV incidence by 90% within 10 years [3]. This approach applies a lesson learned from results of the recently concluded randomized trials in Africa for treatment as prevention to go "beyond universal testing and treatment to universal testing, treatment and prophylaxis to achieve HIV epidemic control" [18].

## Acknowledgments

**Disclaimer:** The findings and conclusions in this article are those of the authors and do not necessarily represent the views of the U.S. Centers for Disease Control and Prevention.

## Author Contributions

**Conceptualization:** Taraz Samandari, Karen W. Hoover, Azfar-e-Alam Siddiqi.

**Data curation:** Ya-Lin A. Huang, Karen W. Hoover, Azfar-e-Alam Siddiqi.

**Formal analysis:** Jeffrey Wiener.

**Investigation:** Taraz Samandari.

**Methodology:** Taraz Samandari, Jeffrey Wiener, Ya-Lin A. Huang, Karen W. Hoover, Azfar-e-Alam Siddiqi.

**Project administration:** Taraz Samandari.

**Resources:** Ya-Lin A. Huang.

**Writing – original draft:** Taraz Samandari, Jeffrey Wiener.

**Writing – review & editing:** Taraz Samandari, Jeffrey Wiener, Ya-Lin A. Huang, Karen W. Hoover, Azfar-e-Alam Siddiqi.

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
