## [Decision Letter · Decision Letter 0]

4 Aug 2020

PONE-D-20-02032

Impact of viral suppression among persons with HIV upon estimated HIV incidence between 2010 and 2015 in the United States

PLOS ONE

Dear Dr. Samandari,

Thank you for submitting your manuscript to PLOS ONE. After careful consideration, we feel that it has merit but does not fully meet PLOS ONE’s publication criteria as it currently stands. Therefore, we invite you to submit a revised version of the manuscript that addresses the points raised during the review process.

The reviewers have suggested some small changes and highlighted points including the potential confounding effects of undiagnosed HIV infections which we ask you to address in your revised manuscript.

We look forward to receiving your revised manuscript.

Kind regards,

Anna C Hearps

Academic Editor

PLOS ONE

Journal Requirements:

2.We note that you have indicated that data from this study are available upon request. PLOS only allows data to be available upon request if there are legal or ethical restrictions on sharing data publicly. For information on unacceptable data access restrictions, please see http://journals.plos.org/plosone/s/data-availability#loc-unacceptable-data-access-restrictions.

3.We note that [Figure(s) 2] in your submission contain [map/satellite] images which may be copyrighted. All PLOS content is published under the Creative Commons Attribution License (CC BY 4.0), which means that the manuscript, images, and Supporting Information files will be freely available online, and any third party is permitted to access, download, copy, distribute, and use these materials in any way, even commercially, with proper attribution. For these reasons, we cannot publish previously copyrighted maps or satellite images created using proprietary data, such as Google software (Google Maps, Street View, and Earth). For more information, see our copyright guidelines: http://journals.plos.org/plosone/s/licenses-and-copyright.

1.    You may seek permission from the original copyright holder of Figure(s) [2] to publish the content specifically under the CC BY 4.0 license. 

Reviewers' comments:

Reviewer's Responses to Questions

**Comments to the Author**

1. Is the manuscript technically sound, and do the data support the conclusions?

Reviewer #1: Yes

Reviewer #2: Yes

2. Has the statistical analysis been performed appropriately and rigorously? 

Reviewer #1: Yes

Reviewer #2: Yes

3. Have the authors made all data underlying the findings in their manuscript fully available?

Reviewer #1: Yes

Reviewer #2: No

4. Is the manuscript presented in an intelligible fashion and written in standard English?

Reviewer #1: Yes

Reviewer #2: Yes

5. Review Comments to the Author

Reviewer #1: General comments:

-This is a novel and important paper. The authors have written a very clear, descriptive, and ecological study with significant contribution to the literature.

-what is “most recent test in each year”? Recent to each end of year follow-up? Were multiple tests (or even cumulative time suppressed) considered?

-All the tables are clear; except table 2 I may suggest grouping (instead of alphabetical), but region so it is easier to see variation between states in each region

-In model, was co-linearity a factor with some of the demographic measures?

-If space allows, may I suggest a comment on ART coverage and testing rates by state in the discussion

-solid discussion + limitations section; very thorough; addressed gaps appropriately

Reviewer #2: Comments to the authors

This paper describes the association between state-level viral suppression among HIV-diagnosed individuals and the fittest HIV incidence rate in the following year across the United States. The authors conclude that increased viral suppression is associated with a modest decrease in HIV incidence in the following year, and report declining trends in HIV incidence across most US states during the study period. This paper describes important population-level evidence to support U=U and Treatment as Prevention, and would be of interest to PLOS-ONE readership. The statistical analysis is sound and appropriate, however there are some major limitations in using viral suppression estimates among HIV-diagnosed individuals as a proxy of community-level viraemia, which should be further discussed.

My overarching concern is that the authors have not accounted for undiagnosed HIV in their analysis. The authors state that viral suppression measures come from data reported to CDC from jurisdictions with complete reporting of HIV viral load tests, so the viral suppression parameter reflects the proportion of *diagnosed* individuals who are virally suppressed. However, the more proximal variable related to HIV incidence would be state-level viraemia, i.e. the proportion of the *entire population* which have detectable HIV. Therefore, large difference in rates of undiagnosed HIV across states / over time may bias results. Are there any estimates of HIV prevalence / rate of undiagnosed HIV in each state which could be used in conjunction with viral suppression rates among those diagnosed to estimate community-level viremia? If this is not possible this should be discussed in the limitations, or the authors should elaborate on the rationale for using viral suppression among those diagnosed as the driver of HIV incidence.

Lines 89-92. The authors state that HIV incidence is estimated using first reported CD4 cell count after HIV diagnosis and a CD4 depletion model. It would be prudent to elaborate on the methods of this model, and the parameters used. Two citations are given, however it is my understanding that each of the paper cited use slightly different methods / parameters in the depletion model. Can the authors provide the model specs or formulae in supplementary materials?

Line112. Can the authors provide a justification for using a 1-year lag period?

Lines 118-119: “Quartiles for estimated HIV incidence rate and percent viral suppression were calculated across the entire time frame”.. do you mean they were calculated for each state for each year? The results “Most states did not experience changes to a lower quartile” suggests this is the case. If so please clarify.. “Annual quartiles for…”

Table 3 please add p-values

Line 193-195: “The impact on HIV incidence rate in Southern states diminished after adjustment for viral suppression and demographic characteristics to being 36% lower compared with the Northeastern states.” This statement is slightly confusing, the impact of what? Suggest changing to something such as “After adjustment for viral suppression and demographic characteristics, HIV incidence rate was 36% lower in Southern states compared to northeastern states” or “The relative difference in HIV incidence in Southern states compared to Northeasterns states diminished after.”

Lines 251-252. “The increases of sexually transmitted diseases during this period in the United States – to the highest recorded levels – suggest that condom use did not increase.29” I think a citation here to actual data on condom use would be more suitable.

Line 264 “In conclusion, our ecologic analysis suggests that between 2010 and 2015, the increase in viral suppression observed after the rollout of ART modestly reduced the estimated HIV incidence rate in the United States.” Given the limitations mentioned and the ecological nature of this study, authors should avoid causal terms, eg reduced. Consider “was associated with a reduction in”.

6. PLOS authors have the option to publish the peer review history of their article (what does this mean?). If published, this will include your full peer review and any attached files.

Reviewer #1: **Yes: **Kate Salters

Reviewer #2: No

---

## [Author Response · Author response to Decision Letter 0]

6 Sep 2020

Journal Requirements:

3.We note that [Figure(s) 2] in your submission contain [map/satellite] images which may be copyrighted. All PLOS content is published under the Creative Commons Attribution License (CC BY 4.0), which means that the manuscript, images, and Supporting Information files will be freely available online, and any third party is permitted to access, download, copy, distribute, and use these materials in any way, even commercially, with proper attribution. For these reasons, we cannot publish previously copyrighted maps or satellite images created using proprietary data, such as Google software (Google Maps, Street View, and Earth). For more information, see our copyright guidelines: http://journals.plos.org/plosone/s/licenses-and-copyright.

AUTHORS: The maps in Figure 2 were created using the urbnmapr package in R. This package uses shapefiles from the US Census Bureau, which are not protected under copyright. 

Reviewers' comments:

Reviewer's Responses to Questions

Comments to the Author

1. Is the manuscript technically sound, and do the data support the conclusions?

Reviewer #1: Yes

Reviewer #2: Yes

2. Has the statistical analysis been performed appropriately and rigorously? 

Reviewer #1: Yes

Reviewer #2: Yes

3. Have the authors made all data underlying the findings in their manuscript fully available?

Reviewer #1: Yes

Reviewer #2: No 

4. Is the manuscript presented in an intelligible fashion and written in standard English?

Reviewer #1: Yes

Reviewer #2: Yes

5. Review Comments to the Author

Reviewer #1: General comments:

-This is a novel and important paper. The authors have written a very clear, descriptive, and ecological study with significant contribution to the literature.

AUTHORS: Thank you.

-what is “most recent test in each year”? Recent to each end of year follow-up? Were multiple tests (or even cumulative time suppressed) considered?

AUTHORS: The most recent test in each year is the last recorded viral load test result within the given calendar year. Multiple tests were not considered, only the last recorded viral load test. Cumulative time suppressed was not considered.

-All the tables are clear; except table 2 I may suggest grouping (instead of alphabetical), but region so it is easier to see variation between states in each region

AUTHORS: This change has been made to Table 2.

-In model, was co-linearity a factor with some of the demographic measures?

AUTHORS: Yes, multicollinearity in the model was a concern since the demographic measures are correlated. However, the estimated rate ratios and precision estimates were similar between unadjusted and adjusted models, indicating validity of the estimated associations was likely not impacted by these correlations.

-If space allows, may I suggest a comment on ART coverage and testing rates by state in the discussion

AUTHORS: Although we reference several articles that studied the impact of ART coverage on HIV incidence (e.g., Montaner PLoS One 2014), we chose to examine the impact of viral suppression on HIV incidence given the well-known phenomenon of non-adherence to ART (Durham Int J STD AIDS 2018, Mills JAMA 2006, Wohl JAIDS 2017). Viral suppression is more proximal to incident HIV and if not achieved will lead to transmission of the virus (Cohen NEJM 2016, Table 2). We have added a sentence to highlight this strength of our analysis in the Discussion. Regarding testing rates, we believe Reviewer #1 is alluding to the concern with underdiagnosis of HIV in the US. We address this in a question raised by Reviewer #2 below and have modified a sentence in the Methods to more explicitly address this concern.

-solid discussion + limitations section; very thorough; addressed gaps appropriately

AUTHORS: Thank you.

Reviewer #2: Comments to the authors

This paper describes the association between state-level viral suppression among HIV-diagnosed individuals and the fittest HIV incidence rate in the following year across the United States. The authors conclude that increased viral suppression is associated with a modest decrease in HIV incidence in the following year, and report declining trends in HIV incidence across most US states during the study period. This paper describes important population-level evidence to support U=U and Treatment as Prevention, and would be of interest to PLOS-ONE readership. The statistical analysis is sound and appropriate, however there are some major limitations in using viral suppression estimates among HIV-diagnosed individuals as a proxy of community-level viraemia, which should be further discussed.

My overarching concern is that the authors have not accounted for undiagnosed HIV in their analysis. The authors state that viral suppression measures come from data reported to CDC from jurisdictions with complete reporting of HIV viral load tests, so the viral suppression parameter reflects the proportion of *diagnosed* individuals who are virally suppressed. However, the more proximal variable related to HIV incidence would be state-level viraemia, i.e. the proportion of the *entire population* which have detectable HIV. Therefore, large difference in rates of undiagnosed HIV across states / over time may bias results. Are there any estimates of HIV prevalence / rate of undiagnosed HIV in each state which could be used in conjunction with viral suppression rates among those diagnosed to estimate community-level viremia? If this is not possible this should be discussed in the limitations, or the authors should elaborate on the rationale for using viral suppression among those diagnosed as the driver of HIV incidence.

AUTHORS: The proportion of PWH virally suppressed for each state was calculated using all persons living with HIV (both diagnosed and undiagnosed) as the denominator, so undiagnosed HIV is accounted for in the calculation of this variable. Our methodology and source data are now more explicitly described in the revised manuscript. 

Additionally, as an alternative, we had repeated the multivariable analysis to include the proportion of diagnosed HIV as a separate state-level variable in the model and the proportion of viral suppression was calculated only for those with diagnosed HIV. The results were similar, with the exception that the effect of region diminished after adjustment for the proportion with diagnosed HIV. These results are included below for comparison.

Unadjusted and adjusted associations of viral suppression (among only those diagnosed with HIV) and demographic variables with HIV incidence

 Unadjusted Rate Ratio (95% CI)* Adjusted Rate Ratio (95% CI)**

Viral Suppression (%) in Previous Year

(per increase of 10%) 1.01 (0.98, 1.04) 0.95 (0.92, 0.98)

Percent Diagnosed with HIV in Previous Year

(per increase of 10%) 1.09 (1.05, 1.14) 1.08 (1.03, 1.14)

Percent < High School Graduate

(per increase of 5%) 1.07 (0.95, 1.21) 1.18 (1.03, 1.35)

Percent Black

(per increase of 5%) 1.34 (1.27, 1.42) 1.36 (1.27, 1.46)

Percent Hispanic

(per increase of 5%) 1.04 (0.89, 1.21) 1.09 (1.03, 1.15)

Percent Uninsured 

(per increase of 5%) 0.93 (0.91, 0.95) 0.94 (0.91, 0.96)

Percent Below Poverty Level

(per increase of 5%) 1.05 (1.02, 1.08) 1.10 (1.06, 1.13)

Percent MSM 2009-13

(per increase of 1%) 1.22 (1.12, 1.33) 1.15 (1.01, 1.30)

Prevalence of HCV Infection 2013-16

(per increase of 1%) 2.83 (1.58, 5.09) 0.99 (0.73, 1.35)

Region 

 Northeast 1.0 1.0

 Midwest 0.90 (0.38, 2.11) 0.92 (0.68, 1.24)

 South 2.48 (1.08, 5.69) 0.81 (0.56, 1.15)

 West 1.05 (0.42, 2.63) 1.00 (0.72, 1.39)

* Results determined using a multivariable mixed-effects Poisson regression models including the variable of interest, a fixed effect for year, and a random intercept.

** Results determined using a multivariable mixed-effects Poisson regression model including all variables in the table, fixed effects for year and 2010 HIV incidence rate, and a random intercept.

Lines 89-92. The authors state that HIV incidence is estimated using first reported CD4 cell count after HIV diagnosis and a CD4 depletion model. It would be prudent to elaborate on the methods of this model, and the parameters used. Two citations are given, however it is my understanding that each of the paper cited use slightly different methods / parameters in the depletion model. Can the authors provide the model specs or formulae in supplementary materials?

AUTHORS: It is beyond the scope of this paper to elaborate the methods of previously published models which our team did not design. While two citations reference the methods, we agree with Reviewer 2 that there is some ambiguity as to the source of the HIV incidence data by state. We have added a reference and modified the Methods section to read as follows: “The National HIV Surveillance System (NHSS) serves as the source for both HIV diagnosis and viral suppression data. CDC estimates and publishes HIV incidence using the reported diagnosis data following the methodology of Song et al. using a CD4 depletion model to estimate time between infection and diagnosis. The estimation method also makes use of first reported CD4 cell count after HIV diagnosis, assuming that persons were ART naïve at the time. Reporting of first CD4 test result after diagnosis is a required element in NHSS and significant progress has been made in reporting of all levels and percentages of CD4 test results. For this analysis, we used CDC’s published HIV incidence estimates among persons 13 years or older (adults and adolescents) during 2010 through 2015.” 

Line112. Can the authors provide a justification for using a 1-year lag period?

AUTHORS: The 1-year lag period was used to most accurately assess the impact of viral suppression on HIV incidence, since viral suppression was defined using the most recent test result in a given year which could often be late in the calendar year. Without a 1-year lag, we would have assumed that all virally suppressed PWH would have been suppressed at the beginning of the year to impact HIV incidence in the same year which seems implausible.

Lines 118-119: “Quartiles for estimated HIV incidence rate and percent viral suppression were calculated across the entire time frame”.. do you mean they were calculated for each state for each year? The results “Most states did not experience changes to a lower quartile” suggests this is the case. If so please clarify.. “Annual quartiles for…”

AUTHORS: To assess trends in viral suppression and HIV incidence over time, quartiles were calculated using all data points from 2010-15, not separately for each year. In the methods we state “Quartiles … were calculated across the entire time frame.”

Table 3 please add p-values

AUTHORS: P-values have been added to Table 3.

Line 193-195: “The impact on HIV incidence rate in Southern states diminished after adjustment for viral suppression and demographic characteristics to being 36% lower compared with the Northeastern states.” This statement is slightly confusing, the impact of what? Suggest changing to something such as “After adjustment for viral suppression and demographic characteristics, HIV incidence rate was 36% lower in Southern states compared to northeastern states” or “The relative difference in HIV incidence in Southern states compared to Northeasterns states diminished after.”

AUTHORS: We agree with Reviewer 2 and have modified the sentence.

Lines 251-252. “The increases of sexually transmitted diseases during this period in the United States – to the highest recorded levels – suggest that condom use did not increase.29” I think a citation here to actual data on condom use would be more suitable.

AUTHORS: We have now provided two references to support this suggestion of a decline in condom use during this period: Paz-Bailey et al. AIDS 2016 found that in the 2005, 2008, 2011, and 2014 cycles of National HIV Behavioral Surveillance: “Among 5371 HIV-positive MSM, there were increases in concordant (19% in 2005 to 25% in 2014, P<0.001) and discordant condomless sex (15 to 19%,P < 0.001). The increases were not different by ART use.” Also, Harper et al. Sex Transm Dis. 2018 who reported that in the 2003–2015 National Youth Risk Behavior Surveys: “Between 2003 and 2015, significant declines in self-reported condom use were observed among black female (63.6% in 2003 to 46.7% in 2015) and white male students (69.0% in 2003 to 58.1% in 2015).”

Line 264 “In conclusion, our ecologic analysis suggests that between 2010 and 2015, the increase in viral suppression observed after the rollout of ART modestly reduced the estimated HIV incidence rate in the United States.” Given the limitations mentioned and the ecological nature of this study, authors should avoid causal terms, eg reduced. Consider “was associated with a reduction in”.

AUTHORS: We agree and the sentence has been modified.

---

## [Editor Report · Decision Letter 1]

2 Oct 2020

Impact of viral suppression among persons with HIV upon estimated HIV incidence between 2010 and 2015 in the United States

PONE-D-20-02032R1

Dear Dr. Samandari,

We’re pleased to inform you that your manuscript has been judged scientifically suitable for publication and will be formally accepted for publication once it meets all outstanding technical requirements.

Kind regards,

Anna C Hearps

Academic Editor

PLOS ONE

---

## [Editor Report · Acceptance letter]

8 Oct 2020

PONE-D-20-02032R1 

Impact of viral suppression among persons with HIV upon estimated HIV incidence between 2010 and 2015 in the United States 

Dear Dr. Samandari:

I'm pleased to inform you that your manuscript has been deemed suitable for publication in PLOS ONE. Congratulations! Your manuscript is now with our production department. 

Kind regards, 

on behalf of

Dr. Anna C Hearps 

Academic Editor

PLOS ONE